# "It's a battle for eyeballs and suicide is clickbait": The media experience of suicide reporting in India

Gregory Armstrong[1]*, Lakshmi Vijayakumar[2,3], Anish V. Cherian[4], Kannan Krishnaswamy[5]

1 Nossal Institute for Global Health, Melbourne School of Population and Global Health, University of Melbourne, Melbourne, Australia, 2 Department of Psychiatry, Voluntary Health Services, Chennai, India, 3 SNEHA Suicide Prevention Centre, Chennai, India, 4 Department of Psychiatric Social Work, National Institute of Mental Health and Neurosciences, Bengaluru, India, 5 George Institute for Global Health, Delhi, India

* g.armstrong@unimelb.edu.au

## Abstract

### Introduction

Suicide rates in India are among the highest in the world, equating to over 200,000 suicide deaths annually. Crime reports of suicide incidents routinely feature in the Indian mass media, with minimal coverage of suicide as a broader public health issue. To supplement our recently published content analysis study, we undertook qualitative interviews to examine media professionals' perspectives and experiences in relation to media reporting of suicide-related news in India.

### Materials and methods

In 2017–18, semi-structured qualitative interviews with twenty-eight print media and television media professionals with experience reporting on suicide-related news were undertaken across north (New Delhi and Chandigarh) and south (Chennai) India. A semi-structured interview guide was designed to initiate discussions around; 1) perspectives on why suicide incidents are regularly reported on by mass media in India, 2) a description of experiences and processes of covering suicide incidents on the crime beat; and 3) perspectives on the emergence of health reporter coverage of suicide. Interviews were digitally audio-recorded and transcribed. A deductive and inductive thematic analytic approach was used, supported by the use of NVivo.

### Results

Suicides were typically seen as being highly newsworthy and of interest to the audience, particularly the suicides of high-status people and those who somewhat matched the middle-class profile of the core audience. Socio-cultural factors played a major role in determining the newsworthiness of a particular incident. The capacity to link a suicide incident to compelling social narratives, potentially detrimental social/policy issues, and placing the

**Data Availability Statement:** All relevant data are within the manuscript. Raw data cannot be shared publicly due to ethical restrictions. Requests for access to the raw data can be sent to the Human

Research Ethics Committee at The University of Melbourne (HREC 1749532.1) at the following email address: HumanEthics-Enquiries@unimelb.edu.au.

**Funding:** The study was funded by an Early Career Researcher Grant from the Society for Mental Health Research. The lead author (GA) holds salary funding in the form of an Early Career Fellowship from the National Health and Medical Research Council (NHMRC) in Australia (GNT1138096).

**Competing interests:** The authors have declared that no competing interests exist.

suicide as a form of protest/martyrdom increased newsworthiness. Reporters on the crime beat worked in close partnership with police to produce routine and simplified incident report-style coverage of suicide incidents, with the process influenced by: informal police contacts supporting the crime beat, the speed of breaking news, extremely tight word limits and a deeply fraught engagement with bereaved family members. It was articulated that a public health and/or mental health framing of suicide was an emerging perspective, which sought to focus more on broader trends and suicide prevention programs rather than individual incidents. Important challenges were identified around the complexity of adopting a mental health framing of suicide, given the perceived pervasive influence of socioeconomic and cultural issues (rather than individual psychopathology) on suicide in India.

## Conclusions

Our findings delve into the complexity of reporting on suicide in India and can be used to support constructive partnerships between media professionals and suicide prevention experts in India. Policymakers need to acknowledge the socio-cultural context of suicide reporting in India when adapting international guidelines for the Indian media.

## 1. Introduction

Suicide rates in India are among the highest in the world. The most recent suicide death rate estimates range between 18 and 21 deaths per 100,000 population (compared with the global average of 11/100 000) [1, 2], equating to an estimated 230 000–250 000 suicide deaths annually. A public health approach to suicide prevention is gaining momentum in India with calls for the development of suicide prevention strategies, including the development of media guidelines to improve mass media coverage of suicides [3].

One of the few successful suicide prevention strategies at the population level is responsible media reporting of suicides [4, 5], based on evidence around copycat suicides, dissemination of suicide methods and behaviours, and the imperative to deliver tailored suicide prevention messaging in media content [6–9]. There are also important concerns based on observations that media can present simplistic monocausal explanations for suicide and can selectively present 'newsworthy' stories that do not reflect the broader array of suicide events in the population, thus impacting the public's knowledge and beliefs about suicide [10].

Consequently, the World Health Organization (WHO) has prescribed guidelines for responsible media reporting of suicide [11]. However, implementation of the guidelines has been varied, and our previous research found a very high frequency of graphic, explicit and simplistic media reporting of suicide in India, predominantly undertaken by crime journalists [12]. We observed that, on average, daily newspapers published an average of one suicide article per day, with the majority being brief incident reports. Potentially harmful reporting practices were common, such as providing a detailed description of the suicide method; potentially helpful practices, such as providing contact details for suicide support services. We also reported some significant disparities between the epidemiological data on suicide in the population and the stories selectively presented for mass media reports [13], indicative of a process whereby media determines which suicides are considered newsworthy. Suicides involving females, younger people aged under 30 and those who were students or farmers were among those groups over-reported relative to their occurrence in the broader population.

The mass media market in India has observed exponential growth and diversification in the number of mass media outlets since the market was privatised in the late 1990s [14]. While other countries have observed a decline in print media, India has maintained steady annual growth in terms of publications and income [15]. Competition is fierce to attract lucrative advertising revenue, with global commercial interests in the huge number of potential consumers in India and the rapidly expanding middle class aspiring to a "Western" lifestyle [16]. As in many countries, this has seen the rapid expansion of a 24/7 breaking news culture. Alongside this trend, the diversity of cultures in India has seen strong demand for a wide array of local/regional news channels, catering to diverse languages and tastes [14]. Consequently, we previously documented that suicide stories in the Indian media are overwhelmingly about incidents that were local to the readership base of the publication, rather than incidents from elsewhere in the country or the world [13].

In late 2019, the Press Council of India issued a press release indicating their adoption of the WHO media guidelines on suicide reporting [17], although India-specific guidelines adapted to the local context are yet to be developed. To help inform future work with media professionals in India, a crucial next step is to better understand the perspectives and experiences of media professionals in relation to reporting on suicide. A review found that there is significant variability in the effect of guidelines on media reporting practices across countries [18]. A key concern raised is that in many countries, it is mental health experts who have largely written the media guidelines without sufficient involvement from media professionals [18]. Journalists can be sceptical about the association between reporting style and imitative suicidal behaviour and may view any restrictions as censorship that can perpetuate "taboos" around suicide [19]. Media experts further emphasise their important role in "agenda-setting", "framing" stories, and "priming" the audience to respond to issues in certain ways, all of which could be tapped into to improve reporting practices [20].

In order to facilitate better engagement with media professionals in India, the aim of this study was to examine their perspectives and experiences in relation to reporting on suicide, including what makes a suicide newsworthy and the processes and challenges of covering suicide events.

## 2. Materials and methods

In early 2018, semi-structured interviews were undertaken with media professionals in India (Chennai, Delhi and Chandigarh) who had previously reported on suicide-related news. Semi-structured interviews were chosen as we knew we would only have the opportunity to do one interview with busy media professionals, and we wanted in-depth data on specific media practices and experiences, Ethics approval was provided in Australia by the Human Research Ethics Committee at The University of Melbourne (HREC 1749532.1). Ethics approval was provided in India by the Institutional Ethics Committee of the Schizophrenia Research Foundation (SCARF) in Chennai. SCARF is a WHO Collaborating Centre for Mental Health Research & Training.

### 2.1 Study setting

Due to resource constraints, we focused on recruiting media professionals working in Chennai and Delhi, where we already had established media connections through our research assistants who were veteran media professionals. Both Chennai and Delhi are in the top 2 cities in India for the highest number of suicide deaths [21], and focusing on these two cities allowed us to examine experiences in both the north and the south of the country. Through the recruitment process in Delhi, we were also placed in contact with two additional participants working

as media professionals in the smaller nearby city of Chandigarh (estimated population ~1 million), a smaller city to the west of Delhi. Tamil is the official language in Chennai and Hindi is the official language in both Delhi and Chandigarh.

## 2.2 Study participants

We interviewed 28 media professionals who had experience reporting on suicide. All types of media professionals working for any type of media outlet were eligible to participate. We used purposive and snowball sampling to recruit participants using a mix of strategies, aiming to have a spread of both vernacular and English-language media outlets and a spread of media professionals working across different media formats. A recruitment advertisement was emailed to major media organisations in New Delhi and Chennai, seeking participation from media professionals who had previously reported on suicide-related news. This was followed-up shortly afterwards with phone calls to these organisations. Snowball sampling was also used, whereby participants identified other potential participants. We also directly approached specific media professionals identified through the networks of our research assistants, who were veteran media professionals.

The sample included a broad diversity of reporters, editors and bureau chiefs, across a wide range of age groups, years of professional experience, and media types (see Table 1). Half the sample primarily reported in English while the other half reported in Tamil or Hindi, with one participant reporting in Telugu. Just over half the respondents (57%) had crime reporting as their primary area of specialisation.

## 2.3 Data collection

Media professionals undertook an audio-recorded face-to-face 45-minute semi-structure qualitative interview with the lead author (GA) to discuss their perspectives and experiences in relation to suicide reporting. Interviews took place at relatively quiet and private locations chosen by the media professionals, which was typically a private room in their workplace, an outdoor setting, or a quiet café/restaurant. Semi-structured interview guides were prepared, in consultation with veteran media professionals, that evolved as the initial interviews proceeded. The guides covered journalists' experiences and perspectives in relation to: 1) what makes a suicide newsworthy, 2) the process and challenges of covering suicide news, and 3) the role of media in suicide prevention (data analysed for a separate manuscript). Interviews were primarily conducted in English, although three participants requested the support of an interpreter. Interviews continued until we had acquired a diversity of media professionals and a sufficient range of responses with no new themes emerging.

## 2.4 Data analysis

A deductive and inductive thematic analytic approach was used [22]. The lead author (GA) transcribed the interviews to enhance familiarisation with the data and prepared an initial code list of major themes based largely around the topic areas discussed in the interview guide. Two coders (GA and AC) read the transcripts multiple times and additional codes were derived inductively for emerging sub-themes, which were driven by the data. The two coders discussed differences in their interpretation of results and refined the coding frame accordingly. The lead author subsequently coded all transcripts, using N-vivo 11 software to organise and manage the data. Throughout the analysis, there were several discussions among the research team, including with veteran media professionals, to ensure accurate interpretation of the data.

**Table 1. Participant characteristics.**

| | n(%) |
|---|---|
| *Gender* | |
| Male | 21 (75.0%) |
| Female | 7 (25.0%) |
| *Age* | |
| 25–30 | 8 (28.6%) |
| 31–40 | 8 (28.6%) |
| 41–50 | 10 (35.7%) |
| 51+ | 2 (7.1%) |
| *Location* | |
| Chennai | 16 (57.1%) |
| Delhi | 10 (35.7%) |
| Chandigarh | 2 (7.1%) |
| *Role* | |
| Reporter | 4 (14.3%) |
| Senior reporter | 13 (46.4%) |
| Editor | 7 (25.0%) |
| Bureau chief | 4 (14.3%) |
| *Years as media professional* | |
| 1–10 | 10 (35.7%) |
| 11–20 | 11 (39.3%) |
| 20+ | 7 (25.0%) |
| *Primary reporting language* | |
| English | 14 (50.0%) |
| Tamil | 9 (32.1%) |
| Hindi | 4 (14.3%) |
| Telugu | 1 (3.6%) |
| *Media type* (multiple responses allowed) | |
| Print newspaper | 17 (60.7%) |
| TV | 12 (42.9%) |
| Online | 12 (42.9%) |
| *Primary content area*[a] | |
| Crime | 16 (57.1%) |
| Health | 8 (28.6%) |
| All content | 4 (14.3%) |

[a] Media professionals, particularly health reporters, tended to have multiple areas of speciality, including in areas like social, political, science and current affairs reporting. Reporters also tended to have moved between areas over time, such as from entertainment reporter to crime reporter. The delineation used in this variable is simply to specify those coming to the interview to discuss their expertise in reporting on suicide from a crime or health perspective.

## 3. Results

We identified three major themes and twelve sub-themes (see Table 2), detailed below.

### 3.1 Newsworthiness

A major area of enquiry was why suicides are so frequently reported on in India, including perspectives and experiences around what makes a suicide event newsworthy. Five main sub-themes emerged.

**Table 2. Coding framework.**

| Master themes | Definition of master theme | Sub themes |
|---|---|---|
| 1. Newsworthiness of suicide in India | Perspectives on why suicide incidents are regularly reported on by mass media in India. What makes suicide newsworthy? Which suicides are more or less newsworthy? | 1. Death makes news in a competitive media landscape<br>2. Compelling social narratives<br>3. Suicide notes and apportioning of blame<br>4. Profile of the deceased<br>5. Simplified reasons for suicide |
| 2. Suicide incidents as crime reports | A description of experiences and processes of covering suicide incidents as routine crime reports. How is information obtained and reports constructed? | 6. Routine crime coverage of suicide<br>7. Police as informants and their relationships with crime reporters<br>8. Time and space limitations<br>9. Contact with bereaved family<br>10. Personal impact of covering suicide |
| 3. Health reporting of suicide | A description of health reporting as a new field in India. Perspectives on the complexities of covering suicide as a public health/mental health issue in India. | 11. Emergence of health reporting on suicide<br>12. Complexities around a mental health framing of suicide |

**3.1.1 'Deaths makes news' in a competitive media landscape.** Participants described how '*crime sells*' [Editor #2, Delhi] and '*deaths make news*' [Senior Reporter #1, Delhi], with deaths and suicide (typically covered by crime reporters) of high interest to the audience. It was articulated that such suicide coverage is often framed as an '*interesting local issue*' [Bureau Chief #1, Delhi], tapping into the audiences' curiosity in unexpected deaths, suicides and other alarming events in their city or neighbourhood. As explained by one news editor:

*Death, as we have been taught in our journalism, is the first casualty. Whenever there is a death, it is newsworthy. It is more sensationalism, infotainment for our readers. Crime sells they say. Crime actually touches our lives. People think it could happen to me also, or my kid could have committed suicide, so it touches your life. Or if it is close to his house he is more involved with it, "that man two blocks from my house has committed suicide, I didn't know that".* [Editor #2, Delhi]

The local crime coverage was also seen as an important part of the metro section of newspapers; it was articulated as the section that helps distinguish newspapers from each other, resulting in a high level of competition for the best local content, as this editor describes:

*The metro section, the team that covers the city events, the crime, including the suicide, is the cutting edge of the newspaper. That is what makes it stand apart from the competitors. . . So, the competition to sensationalise more is greater with crime reporting.* [Editor #2, Delhi]

Participants described how suicide stories attract a lot of interest from the audience, '*Suicide does have the element to engage the audience. . . I think there's a fascination about suicide*' [Editor #3, Delhi]. Those participants who publish news online described suicide content as getting lots of "clicks"; '*the number of online clicks you get is not less than what you get for a sex story or an infertility story*' [Editor #4, Chennai]. The sensationalism associated with coverage of suicide events was characterised as '*a battle for eyeballs. . . suicide is collateral damage, in that sense. . . People are attracted to stories on suicides, it's clickbait*' [Bureau Chief #4, Chennai].

**3.1.2 Compelling social narratives and links to a cultural history of suicide as protest.** All participants emphasised that the newsworthiness of a suicide was intimately linked to the social and political issues surrounding it, and the larger message that can be conveyed to

society. '*Every day we get messages about suicide stories. We only cover suicide stories where we can send a message to society at large.*' [Senior Reporter #8, Chennai]. Reporting suicide events was seen as '*important news because it shows the face of society. Where we are going? How much pressure we can be under? In what condition are we committing suicide?*' [Senior Reporter #2, Delhi]. This senior crime reporter explained that newsworthiness is heavily based on the reason behind the suicide:

> *The reason behind the suicide is what makes it newsworthy. On average, around four to five suicides come to my attention every day, so we pick and choose. Last month, an elderly couple committed suicide because they had been abandoned by their children. In these kinds of cases it is something that we would definitely play. We would say, 'do not abandon your own parents, because they have brought you up'. It's Indian culture, we are supposed to take care of our ageing parents and the family.* [Senior Reporter #11, Chennai]

The majority of participants went further to articulate that reporting suicide incidents became especially newsworthy when it involved highlighting perceived injustices and giving a voice to those who were seen as otherwise voiceless victims of social wrongs. One editor and TV personality explained:

> *I host a daily crime show on TV, half hour each night. We present a minimum of 3–4 suicides each week. Interesting cases, reportable cases. Anything related to injustice, the public think it's their story. Suicide with injustice, the public are very close to such stories and they rate highly. . . There was a classic case. Anita, a Dalit girl and doctor aspirant who committed suicide over an issue with the national medical entrance exams. . . The government suddenly changed the syllabus and she was disadvantaged. . . It played like anything. We covered only the Anita story for 4 or 5 days* [Editor #5, Chennai].

By contrast, suicides thought to be related to mental health reasons were seen as less newsworthy. '*Suicides are two types. One is due to mental illness. The second is the social issue. You can't present just anything uninteresting to the public. If something is a social issue, the public will listen*' [Senior Reporter #7, Chennai].

This motivation to report on suicide incidents to highlight social issues extended to seeing suicide as an opportunity to influence government policy: '*it is only when a suicide happens it's like shock value. . . so that's often why suicides are covered. It's a catalyst for change*' [Reporter #2, Chennai] and '*traditionally, media houses have looked at such suicides as a kind of tool to pressure the government to take action*' [Senior Reporter #9, Chennai]. To affect change, participants described using sensational and sustained media coverage of a suicide event, as outlined by one senior crime reporter:

> *We intentionally sensationalise stories to give prominence to the issue, where it is connected to government policy. . . The suicide is used as a trigger to bring the issue to the forefront. We provide regular updates. We keep tracking the investigation., We ask the political parties for their opinions.* [Senior Reporter #13, Chennai]

Others articulated that they can use sensational media coverage '*so that the government steps in and gives some compensation to the family. We feel good that we have helped in some way*' [Reporter #3, Chennai].

Without prompting, several participants also described beliefs that some instances of suicide in India are a form of protest or martyrdom, a belief they felt had deep socio-cultural

roots. The resonance of this perspective appeared to be especially strong among participants from Chennai, with several participants making reference to historical events where suicide decedents have been memorialised as martyrs. As explained by one senior crime reporter:

*To add to this complex problem, we have a kind of a culture where we glorify such acts. In Tamil Nadu, there is a big history with the anti-Hindi agitations. Lots of people committed suicide. To this day they are being honoured as martyrs, people who died for the Tamil language. Every year, the anniversaries are observed. . . So, this type of culture which supports or encourages suicide in one form or another is a big problem for this society. It isn't just Tamil Nadu. During the Mandal agitation in 1990, protests took place across all of India. The issue was around having reservations or quotas for government jobs to go to people of disadvantaged castes. During the protests, either against or for the policy, I remember so many self-immolation stories from across all of India.* [Senior Reporter #9, Chennai]

For some participants, this was accompanied by a view that, in some circumstances, suicide may be a brave act embodying an important societal message. One senior crime reporter explained:

*It is about the voice of the oppressed. Suicide is voiceless peoples voice. . . For a person to commit suicide, he will be more brave than the average person. It is an act of bravery. . . People want to send a message to the society and the government about their grievances. They use suicide as the message. . . They are martyrs.* [Senior Reporter #7, Chennai]

**3.1.3 Suicide notes and apportioning of blame.** Many participants articulated that suicide notes, written by the deceased prior to their death, were critically important pieces of information that could increase the newsworthiness of a suicide: '*If there is a proper suicide note we go for it*' (Senior Reporter #2). Such notes were seen as crucial in getting an insight into the cause of the suicidal behaviour. '. . .*I got hold of the suicide note and I put it up online. It's the most important part, right? It will explain their thoughts and emotions when they committed suicide*' [Reporter #1, Chennai]. Additional newsworthiness was attained if the suicide note blamed someone or some situation as the cause of the suicidal behaviour, allowing media to generate a level of controversy that would spark reader interest: '*In some cases, someone blames someone for the suicide, sometimes in a suicide note. Such cases are reported bigly. Media reports those cases of suicide where there is a controversy*' [Bureau Chief #2, Delhi]. Typically, suicide notes were seen as trustworthy sources of information, as articulated by one senior crime reporter:

*A business owner committed suicide and wrote a suicide note saying that the demonetisation policy was the reason. So, this made news value. . . Ninety to ninety-five percent of the time, the suicide notes generally reflect the cause for the suicide. There's also a belief in Tamil Nadu that when you're dying, people generally don't lie. . .* [Senior Crime Reporter #12]

**3.1.4 Profile of the deceased.** All participants emphasised that the profile of the suicidal person was critical to the newsworthiness of the story. Typically, there was a view that suicides of lower profile people were of less interest to the audience, who were typically of a middle- or upper-class background and who may feel more connection with the suicide of someone that matches their own profile: '*A well-placed person committing suicide will get more attention from the media. We believe the readers, who are generally educated, will identify themselves with those victims.*' [Senior Crime Reporter #10]. Celebrity suicides in particular were seen to '*have*

*commercial value*' [Editor #5, Chennai]: '*If there is a celebrity involved, then it's definitely page one. Crime, sex and celebrity is page one*' [Editor #4, Chennai].

Meanwhile, a commonly expressed view was that the suicide of someone from a poor background was less newsworthy as it was seen as a less important priority for the country, as articulated below by one news editor:

> *the profile of the victim matters to the treatment that the suicide gets. All suicides are not newsworthy. . . I would not spend resources trying to stop suicides in a slum. I would rather try to save those precious education men who can serve the country more. Because the contribution to the economy from that person from the slum is not as much. So I would rather not spend more on saving his life. It's not a priority for a poor country.* [Editor #2, Delhi]

Participants consistently expressed the view that suicides among younger people and females were generally more newsworthy, as they were presumed to generate greater reader empathy and curiosity. Younger people were seen as more newsworthy because '*he is the future of the country. . . [and] reader psychology is such that they will be more interested in why a young person has committed suicide with their future ahead of them*' [Bureau Chief #1, Delhi]. Most participants struggled to explain why female suicides were considered more newsworthy. A variety of suggestions were made. One senior crime reporter suggested: '*maybe because of sympathy. If there is a women's photo, how she was before, then that is more newsworthy. Female is close to their children. If she had two or three children, who will look after them*' [Senior Reporter #4, Delhi]. Others suggested that '*readers want to read a woman suicide story, it has more masala, more spice*' [Senior Reporter #2, Delhi], and that '*when a lady does it, it is more dramatic and more painful for society to look into*' [Editor #2, Delhi].

Certain occupational categories were also viewed as carrying a higher degree of newsworthiness. For example, the suicides of farmers and students were frequently identified as particularly newsworthy, with a perception that farmer suicides are highly politicised and attract a lot of attention while student suicides cause great concern to many readers who themselves have children or grandchildren at school.

**3.1.5 Simplified reasons for suicide.** In contrast to the narrative that suicides are reported to raise awareness of important social issues, several participants also described how a suicide can become highly newsworthy if it can be framed as having a cause perceived as silly or petty: '*suppose that a 10-year old child is committing suicide just because he couldn't get a good mark. We do some special suicide stories on how many persons have committed suicide just because of petty reasons*' [Senior Reporter #2, Delhi]. One senior reporter recalled that suggesting a suicide was the result of a simplistic reason can capture the attention of the audience:

> *One woman suicided because her husband would not take her to the movies. They were married and only living together for three months. We reported this to say this is a silly reason kill yourself. . . We are at fault for reporting in such a simplistic manner. If you give a silly reason for a suicide, the audience automatically tend to watch, they will also take-it-up online and it can go viral.* [Senior Reporter #8, Chennai]

## 3.2 Suicide incidents as crime reports

The second major thematic area was suicide incidents as crime reports, revealing the experiences and processes of crime reporters covering suicide incidents as routine crime reports. Five main sub-themes emerged.

**3.2.1 Routine crime coverage of suicide.** Participants described how suicide events are typically covered by reporters on the crime beat, and that this practice is deeply embedded: '*Traditionally it is considered a crime-based story. If a suicide happens, the crime reporter will find out the details from the police, suicide has always been reported like that*' [Senior Reporter #1, Delhi] and '*reporting of suicide will never change from this*' [Bureau Chief #1, Delhi].

All participants described the crime report approach to suicide reporting as 'very routine' [Bureau Chief #1, Delhi] incident reporting. Typically, the reporting approach was described as brief and depersonalised: '*For us it is just impersonal information, very dry and unhuman coverage. If you go in-depth, there are a lot of complex issues at play. [But] the suicide reports are impersonal and brief*' [Bureau Chief #2, Delhi]. Participants consistently articulated that the reporting approach tended to follow a basic copy: '*The profile of the person. The apparent reason. Whether there was a suicide note. . . and the method of course. . . so yeah, we are supposed to file a 2 or 3 paragraph story with these basic talking points*' [Senior Reporter #4, Delhi].

**3.2.2 Police as informants and their relationship with crime reporters.** All participants identified that police were almost always the first source of information on a suicide incident: '*It's the local police person who gives the most information. . . We cultivate this in local police and in [police control room]. Not every police person has 'news sense', it is a built-in quality*' [Senior Reporter #2, Delhi]. The criminality of suicide attempt under the Indian Penal Code means that '*every suicide is a [police] case. Everything they have we will ask for, and they will have to tell the information*' [Editor #2, Delhi]. The information would come informally at first, through a police source, followed by a formal request for the official first incident report (FIR). One senior crime reporter explained:

> *There are Whatsapp groups in every police district. So, they usually inform that, you know, so and so incident was reported at this station. The crime reporters, the persons covering the respective beats, are members of these groups. They furnish us with the basic details and photos, often of the bodies or a headshot from when they were alive. The information is reliable, and you can publish that, but of course you call the police station and ask to confirm the details officially.* [Senior Reporter #4, Delhi]

Most participants articulated the relationships between reporters on the crime beat and police as being somewhat symbiotic and interdependent, working together on a daily basis. The examples given ranged from helping police gather information on a case to clear quid pro quo arrangements: '*sometimes we give some money or some alcohol. . . sometimes we do favour and we put their name in an article*' [Senior Reporter #2, Delhi] and '*in many cases, the police get a lot of information from what we find out*' [Reporter #2, Chennai]. In other instances, participants described occasions of police contacting them to encourage them to write a media report about a particular case: '*they sometimes call me up and say this is a really sad case. . . so why don't you write about it*?'. [Editor #7, Chennai]. One Bureau Chief reflected that crime reporters often identify strongly with the police, which impacts the way they approach their work:

> *The first information for the crime reporter on suicides comes from the police version. They take it and reproduce it as it is. And sometimes it's in your face ridiculous. . . We need our crime reporters to ask more critical questions but it's like a Stockholm syndrome. They spend most of their lives in the police station and the commissioner's office, and so they identify with the police.* [Bureau Chief #4, Chennai]

**3.2.3 Time and space limitations.** All participants commented on the very tight space limitations that reporters are working with when covering a suicide incident: '*There's a word limit, 300 or 350 words. Just to describe that event and at least one quote from the police or a neighbour will consume the 350 words*' [Senior Reporter #1, Delhi]. Participants also described the time pressures to report on a suicide quickly: '*Because of competition to break news we need to verify facts quickly. If we get to know the incident now we have to verify it within 15–20 minutes. There is little scope for error and there is little time*' [Senior Reporter #4, Delhi]. Often this reporting was well before official investigations were complete: '*It takes at least 24–48 hours for the autopsy. Media reports usually happen before the autopsy is completed. Sometimes we will give a follow-up story after the autopsy is complete*' [Senior Reporter #9, Chennai]. A higher value appeared to be placed on breaking news quickly rather than getting the details of the suicide story right the first time. This occasionally resulted in problems when the post-mortem findings later contradicted the initial media report, resulting in media outlets needing to '*rectify the story when we get a clearer picture*' [Senior Reporter #10].

**3.2.4 Contact with bereaved family.** There was a drive to capture personal impact narratives and information about the incident from bereaved families to engage the audience. Contact details for family members were often being provided to reporters by the police and all participants identified that engagement with bereaved families shortly after the incident was a fraught and unpleasant part of the process of reporting on suicides. Examples were also provided of interviews undertaken with children shortly after the suicides of parents:

> *In one case, two parents took their life. Who is going to look after the children? I met the children and talked to them. We had some conversation about what happened last night… They were 7 or 8 years old. Total family has been destroyed. They were a high-class family, they had everything.*' [Senior Reporter #2, Delhi]

Participants identified that many families are scared of the '*public shaming*' [Senior Reporter #4, Delhi] and social stigma associated with suicide. Participants described how most families request the media not to cover the case and they sometimes try to conceal the facts. A perceived element of concern for families is how they might be portrayed, with the knowledge that the media may '*report against the family, and other times in favour of the grieving family… the reporting may blame a member of the family for the suicide*' [Senior Reporter # 8, Chennai]. Even though families would request that a suicide is not covered, professionalism and competition was used to justify pursuing the story: '*As a professional, I can't stop any story on the request of a family member. If I do not cover my boss will question me and ask why this other newspaper has covered it and not us*' [Senior Reporter #2, Delhi].

By contrast, in some instances, families wanted the suicide to gain media coverage where they were seeking some form of compensation, where they were desperate for answers as to why their loved one killed themselves, or where they wanted to expose some perceived wrongdoing implicated in the suicide: '*For example, a student committing suicide and the parents want to expose what has happened in the college… then they will want to talk about it so that it doesn't happen to other students in the same institution*' [Reporter #2, Chennai].

**3.2.5 Personal impact of covering suicide.** Participants working on the crime beat discussed having become emotionally disengaged and indifferent to some of the sad suicide incidents that they were covering. It was articulated that such stories were a routine part of their day-to-day professional job, and that they had become emotionally disengaged and indifferent:

> *Have we become insensitive? This is something that we [crime journalists] happen to discuss over drinks sometimes… So now if I get a message at 9pm, at the end of a long day, that a guy*

*has suicided by jumping off a building my first reaction is now 'why the fuck did he have to do this at 8pm in the evening? he could have done it in the morning'. I'm not meaning to be insensitive; I'm just being frank. Each day we have to write 6 or 7 reports for different crimes. . . No one taught us to emotionally disengage to the stories, but it is probably a normal human response. The thing is, I used to be a naïve person, a very sensitive person. I don't know how I ended up being a crime reporter and becoming so indifferent.* [Reporter #4, Chennai]

However, in contrast to this notion of having become emotionally numb to such incidents, most participants did report that 'sometimes it is very emotional' [Editor #5, Chennai] and that they occasionally experienced distress and emotional involvement in the suicide report. Participants described feeling disturbed, guilty, angry, shaken, and scared: '*Sometimes the incidents are very bone chilling and they affect you mentally and psychologically. . . There was one time I couldn't sleep for two or three days because I was scared of sleeping alone after covering a suicide*' [Senior Reporter #4, Delhi].

### 3.3 Health reporting on suicide

The third major thematic area was the emergence of health reporting on suicide, revealing the relatively new approach of health reporters covering suicide as a public health issue. Two main sub-themes emerged.

**3.3.1 Emergence of health reporting on suicide.** In strong contrast to the routine incident-report approach of crime reporters, many health reporters and editorial-level media professionals talked about the recent shift to introducing some coverage on suicide as a public health issue. The health reporter angle involved looking at trends and broader social and psychological causes, including bringing in discussion around depression, tension, mental health policies and programs and the prevention of suicide:

*The health reporter's perspective is fairly recent in India. Traditionally it is considered a crime-based story. . . In the last five years, we have also started reporting on suicide from a public health perspective. More people are trying to analyse it and find out the trends and the causes, this is mostly only happening in the English-language media, not the vernacular media. Crime journalists will still report the individual cases.* [Senior Reporter #1, Delhi]

Some health reporters lamented the sensationalism of the dominant crime reporter approach to the coverage of suicide news:

*I'm not reporting on the fire, I'm reporting on what caused the fire. That's what I wish more journalists would do. We're all seeking the sensation that telling you about a suicide will get. I find that very disappointing. The way I understand my profession as a health reporter, it is to seek the broader reasons why these things are happening.* [Editor #3, Delhi]

**3.3.2 Complexities around a mental health framing of suicide.** Many participants talked about how most suicide reporting in India does not draw links with mental illness as a contributing factor to suicide. Depression and its link to suicide was identified as '*something that has been missing from our coverage; I don't think it's something we really understand, it is a new conversation*' [Editor #3, Delhi]. Almost all reporters identified that they themselves had limited mental health literacy and that there is a high stigma attached to mental illness that restricts help-seeking and prevents more open reporting on the topic. As articulated by one editor:

*In India, depression is a silent killer, it is not treated as a disease. People [with depression] have been branded as mentally sick or weak, so they don't report and are not treated. To be treated for mental sickness is a big taboo in India. As a health reporter, I have written features, you know, there are psychiatrists who have packages of care. But the individual, because of social stigma, people don't access help.* [Editor #2]

Stigmatising attitudes towards suicide was also perceived to be a barrier to writing stories around people who are recovering from suicidal thoughts and behaviour. Similarly, the lack of supply of high-quality mental health professionals was also identified as a barrier to any positive gains that might otherwise emerge from stimulating awareness of mental health services:

*suicidal thoughts are actually quite common, and we need more personal stories on this in the media, but for the person sharing their story it puts them under a lot of pressure to say this in public. . . A lot of people don't trust counselling. People think they are a waste of time. I think they can help, if you are willing to trust them and they are good. I think the worst possible thing is when you finally go to get help and you get a terrible counsellor or you can't access a service. We need more and better counsellors.* [Reporter #1, Chennai]

Some participants also raised issues around the sociocultural and economic complexity of suicide in India, articulating that structural drivers of suicide are prominent with people facing real and multifaceted challenges that impact on suicide rates. Concerns were raised that shifting to a narrow focus on mental health as a key driver of suicide in India would not do justice to the range of complex societal issues at play:

*Suicide raises so many issues. The way depression has been ignored, we've also ignored a lot of the social and economic issues surrounding suicide. . . Like say, the disempowerment of women or a homeless man dying. We should be equally responsible for those deaths to. . . These are human beings who are facing complex problems. Depression is only talked about when it is a well-to-do person. I happened to cover one report of a young women who had attempted suicide. . . I'm sure depression was an issue, but so was hunger and poverty.* [Editor #3, Delhi]

Finally, it was also articulated that it was hard to generate demand for more writing around suicide prevention, in part because the country's economic challenges and the enormous population size have meant that preventing suicide hasn't been a major priority:

*Suicide in India is not shaking the social concerns as much as it does in the West. We are over-populated. One more death doesn't really shake up people, unless it's a high-profile thing. In India, we haven't reached the point of focusing so much on preventing suicide. . . To us, it is just one more death, it is in the hands of the Gods.* [Editor #2, Delhi]

## 4. Discussion

One of the few successful suicide prevention strategies at the population level is responsible media reporting of suicides [4, 5], yet this issue has remained under-examined in the Indian context. In order to facilitate better engagement with media professionals in India on this issue, we sought to examine their perspectives and experiences in relation to suicide reporting. Our findings reveal the perspectives and experiences of media professionals in India regarding the newsworthiness of suicide and the processes and challenges of covering suicide news.

Commercial competitiveness was identified as a key motivating factor for reporting on suicide events, reflecting similar findings in Hong Kong and New Zealand [19, 23]. Our participants generally articulated that the high frequency of suicide reporting was in-part driven by a perception that suicide can be highly newsworthy and of interest to the audience, justified by anecdotal reports of internal data showing that online suicide stories were generating a large number of "clicks".

Our findings highlighted that socio-culture issues also play a major role in determining the newsworthiness of suicide events in India. Compelling social narratives that are based in cultural mores were believed to stimulate greater audience interest in a suicide. Furthermore, the capacity to link a suicide event to the detrimental impact of social/policy issues allowed a suicide report to generate controversy and apportioning of blame, again believed to stimulate audience interest. For some media professionals, this was linked with beliefs in India that some suicide events are a form of protest and martyrdom related to larger social issues, informed by major historical events where protest suicides took place. The concept of suicide as protest or martyrdom is a phenomenon that has also appeared in the academic literature on suicide in India, especially in reference to suicide by self-immolation [24, 25]. In connection with this, media professionals see themselves as playing an important advocacy role in using suicides as a catalyst for raising awareness about social issues. A similar finding was observed in Hong Kong [23], leading the authors to assert that suicide is perceived to be the result of social problems in many Asian societies. In one sense, this resonates with the push for a taking a broad public health approach to suicide prevention that equally emphasises the role of major social issues and policies rather than solely focusing on individual-level pathologized distress [26]. In another sense, such a predilection may lead to an over-simplification of suicide, including overlooking the important role of individual-level suicide risk factors like mental illness and drug and alcohol issues. Importantly, it is worth considering the extent to which reporting on suicide from such a framework may risk justifying the suicidal behaviour, inadvertently role-modelling suicide as an acceptable option in the face of intractable social ills. It may also role-model suicidal behaviour as a way to gain public notoriety via media coverage and to seek public retribution against people or institutions.

The perceived newsworthiness of a suicide event was heavily influenced by the profile of the suicidal person, as found in other studies of media professionals elsewhere in the world [19, 23, 27]. Our participants ascribed greater newsworthiness to suicides of youths, women, certain occupational categories like farmers and students, which is highly consistent with our previous comparison of mass media reports of suicides compared to the epidemiological data [13]. Our participants also articulated a preference for stories on suicides of people whose profile matches the educated middle-class news audience, to increase the chance that the audience will identify with them. Such preferences have also been observed elsewhere. For example, in New York media professionals sought to identify suicides that increased the chance that the audience might identify with the suicidal person [27], and a preference for youth suicides has been observed in Austria, Australia and China [28–30]. While based on a desire to capture a greater market share, this approach to selecting newsworthy suicides for publication also carries several risks to the population. Firstly, the selection biases may cause the general public to develop misunderstandings about who is impacted by suicide and may influence political priority-setting around suicide prevention policies and programs. Secondly, the effect of the media on copycat suicidal behaviour can be pronounced when the audience identifies and comes from the same demographic group as the model portrayed in the media stimulus [31], described as a *horizontal identification* phenomenon [32], raising concerns about the preference for reporting on suicides of people who are a demographic match for the audience. Similarly, our participants' articulated preference for higher class and celebrity suicide stories

carries risks of *vertical identification*, whereby some audience members may be influenced by the suicidal behaviour of "superior" models [33].

The newsworthiness attributed to suicides for seemingly simplistic or petty reasons is also concerning. International literature on suicide heavily emphasises how self-harm and suicide are never the result of a single factor. Explanations for suicide are multiple, complex and occurring at different levels and time periods, encompassing a range of psychiatric, psychological, cultural and socio-economic risk factors [34]. As such, a key media recommendation is to avoid presenting simplistic monocausal explanations for suicide [11]. Relatedly, the heavy emphasis our participants placed on suicide notes as reliable sources of information on the reason for the suicide was equally concerning, given such notes are likely to emphasise recent social and interpersonal stressors rather than a more complex assessment of medium and longer-term factors and individual-level issues like alcoholism and mental illness.

It's clear that both reporters on the crime beat and the police are playing dominant roles in the production of suicide news in India, with police potentially being a key stakeholder in improving reporting practices. The crime reporter approach to suicide news was described as largely incident based, following a simple format for writing a news story, answering 'who, what, why, where and when' [35], with information and photos provided readily by police officers. Despite the decriminalisation of suicide in India, there appears to be a *street-level bureaucracy* phenomenon at play. Street-level bureaucracy theory [36, 37] seeks to explain the working practices of people delivering services on the frontline and how it may differ from public policy. While the national approach has recently shifted away from criminality of suicide towards a public health framework, it is still "business as usual" in terms of the crime coverage of suicide news. Police are the unofficial and the official sources and they are sharing the personal and sensitive information of those who have killed themselves with crime reporters who have no knowledge or training in responsible approaches to covering suicide news. The reports are occurring before post-mortems have been concluded, with the emphasis on speed of breaking news rather than accuracy. The high frequency of reporting and the complex emotions associated with suicide also impacted the media professionals, many of whom identified instances of personal distress. Participants used professionalism, black humour and cynicism to mediate this distress, a phenomenon also observed among media professionals in New Zealand [19].

The interviews highlighted that media engagement with bereaved families was fraught, with many families pleading for privacy to avoid public shaming. From the accounts of the media professionals we interviewed, it appears that bereaved families and the deceased have severely restricted rights to privacy with their contact and other details being distributed by the police, and with media seeking to interview family, including young children, at a time of complicated grief. An historical criminality framework around suicide coupled with notions of professionalism and commercial competition appear to be combining to justify an approach of overlooking the sensitivities for bereaved families. These findings indicate an important need to develop guidelines for engaging with families bereaved by suicide, as per international recommendations [11]. Further research is clearly needed with bereaved families to document their experiences with media professionals, to better understand this phenomenon and how media can best engage with those bereaved by suicide.

It is clear that a public health framework around suicide reporting is emerging in India, with health reporters starting to tackle stories on the trends and causes of suicide, including the role of poor mental health and approaches to preventing suicide. However, media professionals held concerns that it is not possible to simply replicate the mental health discourse used in 'the West' when discussing suicide in India. This emerging public health discourse needs to be able to equally incorporate the cultural and socio-economic drivers of suicide in

India [38, 39]. Important concerns were raised that it may not be appropriate for the Indian media to start promoting mental illness framings and treatments for suicide, when in many cases it may be understandable (rather than pathological) human distress in the face of serious social issues, such as gender inequality and caste-discrimination. Kleinman's theory of social suffering has some relevance to this discussion [40]. Kleinman uses the term *social suffering*: 1) to emphasise that health problems are at times caused by socioeconomic, cultural and political forces, such as the link between depression and poverty; and 2) to break down the distinction between health and social problems, and framing ill-health as demanding both health and social responses. This perspective may be helpful in shaping nuanced media framings of suicide in India.

We adopted a rigorous and systematic approach to this study. One author conducted and transcribed all interviews, two authors independently undertook coding, and all authors contributed to the final coding frame and interpretation of data. The deductive and inductive approach to thematic analysis allowed for both a comprehensive focus on some key areas of enquiry and the emergence of a wide range of sub-themes. The sample included 28 media professionals from two major cities and one smaller city across print, TV and online news mediums. One-quarter of participants were operating at an editorial level and four operating as Bureau Chiefs, holding a high level of knowledge of reporting practices at their media houses. Nonetheless, there were some limitations. Our findings may not reflect the situation outside of Delhi and Chennai, including non-urban settings. Participants with a special or passionate interest in this issue may have been more likely to offer their participation and we may have missed the perspectives of those media professionals who are more disinterested in the issue. Finally, the interviews were largely conducted in English rather than in participants' first languages, which may have impacted on the quality of the data.

Despite the above-mentioned limitations, we believe there are some important implications from our findings. Firstly, while the move by the Press Council of India to adopt the WHO media guidelines for suicide reporting is a wonderful first step, there would be value in creating India-specific guidelines that are relevant to the cultural context within which the media professionals are operating. Such a guidelines document would also provide contextually relevant examples of preferred reporting approaches, as well as guidance in culturally sensitive areas like how to communicate with bereaved family members. Engaging people with lived experience will be a critical addition to the process. There is also clearly an imperative to use this process to seek agreement on guidelines across the prominent media houses, given competition for breaking news appears to be implicated in the creation of sensational suicide coverage. Secondly, India-specific guidelines would benefit from equally embracing the socio-cultural and environmental drivers of suicide in India, alongside the individualist-oriented mental health narrative, which appears to be insufficient on its own in this context (if not all contexts). Emphasis needs to be given to the multi-factorial and complex pathways to suicide to add more nuance to discussions around suicide in the Indian media. This will support media professionals to avoid simplistic and sensational suicide narratives and to better educate the public about the causes of suicide and the range of suicide prevention options to be considered. Finally, targeted efforts are needed to engage police and crime beat journalists specifically on this topic, as they are key stakeholders in the production of suicide content in the Indian media. The emerging breed of health journalists will be critical allies in all these endeavours.

## 5. Conclusions

As an international standard, the WHO media reporting guidelines play an important role in stimulating interest in improving media reporting of suicide news across the globe. However,

in order to support the development of adapted country-specific guidelines it is essential to examine the richness of the perspectives and experiences of local media professionals. Our findings delve into the complexity of reporting on suicide in India and can be used to support constructive partnerships between media professionals and suicide prevention experts in India.

## Acknowledgments

We sincerely thank all the study participants and Ms Sandhya Ravishankar who supported the recruitment of participants alongside Mr Kannan Krishnaswamy.

## Author Contributions

**Conceptualization:** Gregory Armstrong, Lakshmi Vijayakumar, Kannan Krishnaswamy.

**Data curation:** Gregory Armstrong.

**Formal analysis:** Gregory Armstrong, Lakshmi Vijayakumar, Anish V. Cherian.

**Funding acquisition:** Gregory Armstrong, Lakshmi Vijayakumar.

**Investigation:** Gregory Armstrong, Lakshmi Vijayakumar, Kannan Krishnaswamy.

**Methodology:** Gregory Armstrong, Lakshmi Vijayakumar.

**Project administration:** Gregory Armstrong.

**Resources:** Gregory Armstrong, Kannan Krishnaswamy.

**Software:** Gregory Armstrong.

**Supervision:** Gregory Armstrong.

**Validation:** Kannan Krishnaswamy.

**Visualization:** Gregory Armstrong.

**Writing – original draft:** Gregory Armstrong, Lakshmi Vijayakumar, Anish V. Cherian.

**Writing – review & editing:** Gregory Armstrong, Lakshmi Vijayakumar, Anish V. Cherian, Kannan Krishnaswamy.

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
