## [Decision Letter · Decision Letter 0]

12 Jun 2020

PONE-D-20-10924

“It’s a battle for eyeballs and suicide is clickbait”: the media experience of suicide reporting in India

PLOS ONE

Dear Dr. Armstrong,

Thank you for submitting your manuscript to PLOS ONE. After careful consideration, we feel that it has merit but does not fully meet PLOS ONE’s publication criteria as it currently stands. Therefore, we invite you to submit a revised version of the manuscript that addresses the points raised during the review process.

We look forward to receiving your revised manuscript.

Kind regards,

Conor Gilligan

Academic Editor

PLOS ONE

Journal Requirements:

1.Please ensure that your manuscript meets PLOS ONE's style requirements, including those for file naming. The PLOS ONE style templates can be found at https://journals.plos.org/plosone/s/file?id=wjVg/PLOSOne_formatting_sample_main_body.pdf and https://journals.plos.org/plosone/s/file?id=ba62/PLOSOne_formatting_sample_title_authors_affiliations.pdf

2. Please include a copy of the topic/interview guide used in the study, in both the original language and English, as Supporting Information, or include a citation if it has been published previously.

3.We note that you have indicated that data from this study are available upon request. PLOS only allows data to be available upon request if there are legal or ethical restrictions on sharing data publicly. For information on unacceptable data access restrictions, please see http://journals.plos.org/plosone/s/data-availability#loc-unacceptable-data-access-restrictions.

Additional Editor Comments (if provided):

Thank you for this interesting and worthwhile paper. As you will see, both reviewers were positive about your paper and have recommended only minor revisions. I add my suggestions to theirs, in particular:

- please provide some more description of the socio-cultural (and legal) context of suicide in India in the introduction

- would it be possible to articulate more defined recommendations resulting from this work?

Reviewers' comments:

Reviewer's Responses to Questions

**Comments to the Author**

1. Is the manuscript technically sound, and do the data support the conclusions?

Reviewer #1: Yes

Reviewer #2: Yes

2. Has the statistical analysis been performed appropriately and rigorously? 

Reviewer #1: Yes

Reviewer #2: Yes

3. Have the authors made all data underlying the findings in their manuscript fully available?

Reviewer #1: No

Reviewer #2: No

4. Is the manuscript presented in an intelligible fashion and written in standard English?

Reviewer #1: Yes

Reviewer #2: Yes

5. Review Comments to the Author

Reviewer #1: The study sought to examine media professionals' perspectives and experiences in relation to media reporting of suicide-related news in India using a qualitative approach. 28 print and television media professionals with experience reporting on suicide-related news were interviewed across India by dividing the country into north and south and selecting 2 communities within the divisions based on feasibility and suicide rates. The interview discussions were around: 1) perspectives on why suicide incidents are regularly reported on by mass; 2) a description of experiences and processes of covering suicide incidents on the crime beat; and 3) perspectives on the emergence of health reporter coverage of suicide. A deductive and inductive thematic analysis was used to interpret the data. The study concludes among others that socio-cultural factors played a major role in determining newsworthiness; suicides were seen as highly newsworthy, particularly that of high-status people; linking a suicide incident to compelling social narratives and suicide as a form of protest/martyrdom increased newsworthiness.

Overall, the study answers the questions it seeks to address and uses a good approach to address the questions. The paper will add knowledge to the existing literature on media reporting of suicide and the way forward in India.

Specific Issues to be addressed

1. Introduction

Concise and relevant knowledge of suicide in the India context, however, a brief information on the news culture generally in India together with the guidelines presented in this paper and perhaps how these guidelines have shaped the media landscape will help to situate and better understand the experiences and perceptions held by the informants.

2. Methods

• The authors have attempted to state the issue of trustworthiness of their methodology and data gathering process such as inter-rater coding. However, the authors can further clarify and describe the audit trail process. It is important for authors to address how their research findings are credible, transferable, confirmable, and dependable.

• It will also be helpful if the coding systems and interpretation processes are clearly described and perhaps a bit more explanation on how the categories/themes were derived from the data. It is important that the authors state that the themes identified were data-driven and not due to pre-existing knowledge of the themes.

• The authors have not stated where the interviews were conducted, was it at their respective workplaces, the researcher's office, etc.

• Authors could clarify further this statement "We also directly approached specific media professionals identified through the networks of our research assistants, who were veteran media professionals". How exactly did the research assistant use their networks?

• Although the authors used semi-structured interviews which implied qualitative study, it appears they haven't justified the rationale for choosing the methods in the introduction and methods section.

3. Discussion

• From the narratives, contact for families bereaved is obtained through the police. While breach of confidentiality issues and challenges as a result can be inferred from that, one cannot conclude on the specific effect on families as a result as this was not directly explored in the current study. I would therefore recommend it is left out of the discussion.

• A potential critique for the discussion is the failure to mention any helpful features recommended by WHO or point readers to where they can see help like the health services.

• Implications of the study are not clearly defined, authors should address this.

• An observation is that authors describe media reportage as a suicide prevention strategy in the introduction but no mention of this in the discussion.

Reviewer #2: Review for manuscript entitled ----“It’s a battle for eyeballs and suicide is clickbait” the media experience of suicide reporting in India.

This is an interesting qualitative study interviewing reporters in India to understand what makes a suicide newsworthy and the process of covering suicide news. The information collected is new and important.

Specific comments:

1. What was the proportion of suicide events being reporting India? What is the pattern of suicide news reporting in India? Is there any empirical evidence on the association between media reporting and suicide incidence in India? Contextualizing suicide reportage in India can provide readers with some background information (this should be provided in the introduction).

2. Similarly in Discussion section, explanations should consider socio-cultural background in India. Suicide events are oftentimes treated in the India media as social ills, why is that? Is it related to the long history of criminalization of attempted suicide? Or social inequality (e.g. caste system) ? The 2nd paragraph of discussion should go over and above “suicide is perceived to be the result of social problems in many Asian societies”, specific socio-cultural issues that is related to suicide news reporting in India should be raised and discussed.

3. Any policy implication for the current findings?

6. PLOS authors have the option to publish the peer review history of their article (what does this mean?). If published, this will include your full peer review and any attached files.

Reviewer #1: No

Reviewer #2: No

---

## [Author Response · Author response to Decision Letter 0]

27 Aug 2020

Reviewer #1

“The study sought to examine media professionals' perspectives and experiences in relation to media reporting of suicide-related news in India using a qualitative approach. 28 print and television media professionals with experience reporting on suicide-related news were interviewed across India by dividing the country into north and south and selecting 2 communities within the divisions based on feasibility and suicide rates. The interview discussions were around: 1) perspectives on why suicide incidents are regularly reported on by mass; 2) a description of experiences and processes of covering suicide incidents on the crime beat; and 3) perspectives on the emergence of health reporter coverage of suicide. A deductive and inductive thematic analysis was used to interpret the data. The study concludes among others that socio-cultural factors played a major role in determining newsworthiness; suicides were seen as highly newsworthy, particularly that of high-status people; linking a suicide incident to compelling social narratives and suicide as a form of protest/martyrdom increased newsworthiness. Overall, the study answers the questions it seeks to address and uses a good approach to address the questions. The paper will add knowledge to the existing literature on media reporting of suicide and the way forward in India.”

Thank you for these encouraging comments.

“Concise and relevant knowledge of suicide in the India context, however, a brief information on the news culture generally in India together with the guidelines presented in this paper and perhaps how these guidelines have shaped the media landscape will help to situate and better understand the experiences and perceptions held by the informants.”

Thank you for this suggestion. We have incorporated an additional paragraph to provide some brief context on the mass media sector in India. 

Regarding the second part of this comment, we do not actually provide guidelines in this paper. Rather, we seek to better understand the experiences and perspectives of media professionals as per our objectives. As we state in the paper, the Press Council of India has adopted the World Health Organization media guidelines for suicide reporting, but this was very recent (in late 2019), and we haven’t yet had time to see how these may help to shape (or not) the media landscape. The information in this manuscript will be crucial in efforts to develop contextually relevant India-specific guidelines, co-designed with media professionals. Further, it will assist us to shape appropriate training interventions that address the relevant concerns held by media professionals. 

“The authors have attempted to state the issue of trustworthiness of their methodology and data gathering process such as inter-rater coding. However, the authors can further clarify and describe the audit trail process. It is important for authors to address how their research findings are credible, transferable, confirmable, and dependable.”

It was not an inter-rater coding process as one might use in quantitative analysis. Indeed, we have used such processes before, and it is not relevant on this occasion. In qualitative research, one can never 100% guarantee reproducible findings as the perspective of the researchers will influence the interviews and their interpretation of the findings. It is essentially an interpretive process and there is plenty of literature about this. It is widely accepted that qualitative research, while challenging in terms of reproducibility, is essential to gather in-depth interpretations of social phenomena. 

Nonetheless, we obviously aimed for rigor. We followed standard practice for analysing semi-structured qualitative interview data, whereby two coders (rather than one coder) independently read the transcripts and prepared their thoughts regarding coding, which were then discussed and formulated into a refined coding frame. We believe this process is adequately described in the methods.

“It will also be helpful if the coding systems and interpretation processes are clearly described and perhaps a bit more explanation on how the categories/themes were derived from the data. It is important that the authors state that the themes identified were data-driven and not due to pre-existing knowledge of the themes.”

We have stated in the data analysis section that we used a mix of deductive and inductive coding, with the major themes largely deductively coded and the sub-themes inductively coded (i.e. data driven). The initial deductive coding is because it was a semi-structured interview process with specific themes we were wanting to investigate. All sub-themes related to this were inductively coded by two coders. We believe this process is described adequately in the paper, and we have added that this was a data driven process particularly for the inductive coding.

“The authors have not stated where the interviews were conducted, was it at their respective workplaces, the researcher's office, etc.”

We now provide this information in the methods section, under ‘data collection’.

“Authors could clarify further this statement "We also directly approached specific media professionals identified through the networks of our research assistants, who were veteran media professionals". How exactly did the research assistant use their networks?”

The research assistants, who were veteran media professionals, assisted in identifying a few health and crime journalists who they knew had (or were known by their contacts to have) experience covering suicide news. As mentioned in the methods, it was a multi-pronged approach using both a recruitment email, snowball sampling, and using the networks of our research assistants.

“Although the authors used semi-structured interviews which implied qualitative study, it appears they haven't justified the rationale for choosing the methods in the introduction and methods section.”

We now provide this information at the beginning of the methods section.

“From the narratives, contact for families bereaved is obtained through the police. While breach of confidentiality issues and challenges as a result can be inferred from that, one cannot conclude on the specific effect on families as a result as this was not directly explored in the current study. I would therefore recommend it is left out of the discussion.”

We appreciate the remark by the reviewer; however, we feel this text is fundamentally important to retain in the discussion section. It is rare data indicating that in the experience of media professionals, bereaved families often do not want media coverage of the suicide deaths of their relatives. This is profoundly important, and ought to be reflected on when discussing the development of media guidelines developed specifically for the Indian context, including (as we state in the relevant paragraph) the development of guidelines specifically related to engaging with bereaved families. 

We feel it will help if we add a sentence at the end of the paragraph saying “Further research is clearly needed with bereaved families to document their experiences with media professionals, to better understand this phenomenon and how media can best engage with those bereaved by suicide.”

“A potential critique for the discussion is the failure to mention any helpful features recommended by WHO or point readers to where they can see help like the health services.”

Thank you for this remark. We do appreciate this concern. However, this paper is not intended to be a description of the recommended approaches to reporting. We argue that it is a worthwhile endeavour to focus the paper on the experiences and perspectives of the media professionals themselves. This paper represents a time for public health professionals to listen and reflect, rather than preach (so to speak). The paper provides a reference to the WHO guidelines, which is freely available to all. It is not the role of this paper to provide information about health services, particularly as the health system options for suicide prevention vary dramatically across India and the world (with many readers likely coming from outside India).

“Implications of the study are not clearly defined, authors should address this.”

We have added a new paragraph in the discussion section where we draw out the key implications of the findings.

“An observation is that authors describe media reportage as a suicide prevention strategy in the introduction but no mention of this in the discussion.”

Throughout the discussion we have highlighted the implications of the findings for suicide prevention. We have now added an implications paragraph in the discussion and a new sentence at the beginning of the discussion section to bring the focus back to the role of media in suicide prevention. We also inform the reviewer that we will be publishing an additional paper from these interviews specifically addressing the perspectives of media professionals on the role of media in suicide prevention. That article will spend a lot more time discussing media’s role in supporting suicide prevention, whereas this article focuses on the way suicide stories are currently constructed and what makes a suicide story newsworthy. 

Reviewer #2 

“This is an interesting qualitative study interviewing reporters in India to understand what makes a suicide newsworthy and the process of covering suicide news. The information collected is new and important.”

Thank you for these encouraging comments. 

“What was the proportion of suicide events being reporting India? What is the pattern of suicide news reporting in India? Is there any empirical evidence on the association between media reporting and suicide incidence in India? Contextualizing suicide reportage in India can provide readers with some background information (this should be provided in the introduction).”

Thank you for this comment. We have added additional text in the introduction section where we discuss our previous findings on the pattern of suicide news reporting in India.

Just to inform the reviewer, at present there is no study on the association between media reporting and suicide incidence in India (so we currently rely on evidence from elsewhere). The key limitation being the lack of individual-level suicide data collected. The official data, collated by the National Crime Records Bureau, is kept in aggregate form and is unsuitable for this task. We are currently looking at other options to examine this phenomenon in India and hope to add to the literature shortly. 

“Similarly in Discussion section, explanations should consider socio-cultural background in India. Suicide events are oftentimes treated in the India media as social ills, why is that? Is it related to the long history of criminalization of attempted suicide? Or social inequality (e.g. caste system) ? The 2nd paragraph of discussion should go over and above “suicide is perceived to be the result of social problems in many Asian societies”, specific socio-cultural issues that is related to suicide news reporting in India should be raised and discussed.”

Thank you for this interesting comment. We can’t definitively answer the question as to why suicide events are treated like social ills in the Indian media. Nonetheless, we have used the discussion section to highlight the socio-cultural issues related to the experiences and perspectives offered by the media professionals we interviewed, such as suicide stigma and shame, the criminality ascribed to suicide in India (until recently), and the concept of suicide as protest or martyrdom. We agree that this issue requires greater insights from historians and sociologists to help unpack these important socio-cultural issues around suicide.

“Any policy implication for the current findings?”

We have added a new paragraph in the discussion section where we draw out the key implications of the findings.

---

## [Editor Report · Decision Letter 1]

3 Sep 2020

“It’s a battle for eyeballs and suicide is clickbait”: the media experience of suicide reporting in India

PONE-D-20-10924R1

Dear Dr. Armstrong,

We’re pleased to inform you that your manuscript has been judged scientifically suitable for publication and will be formally accepted for publication once it meets all outstanding technical requirements.

Kind regards,

Conor Gilligan

Academic Editor

PLOS ONE

---

## [Editor Report · Acceptance letter]

8 Sep 2020

PONE-D-20-10924R1 

“It’s a battle for eyeballs and suicide is clickbait”: the media experience of suicide reporting in India

Dear Dr. Armstrong:

I'm pleased to inform you that your manuscript has been deemed suitable for publication in PLOS ONE. Congratulations! Your manuscript is now with our production department. 

Kind regards, 

on behalf of

Dr. Conor Gilligan 

Academic Editor

PLOS ONE